# The Role of Adenogenesis Factors in the Pathogenesis of Endometriosis

**DOI:** 10.3390/ijms26052076

**Published:** 2025-02-27

**Authors:** Pietro G. Signorile, Alfonso Baldi, Rosa Viceconte, Mariarosaria Boccellino

**Affiliations:** 1Italian Endometriosis Foundation, Formello (RM), 00060 Rome, Italy; aie@endometriosi.it (R.V.); m.boccellino@unilink.it (M.B.); 2Department of Life Science, Health and Health Professions, Link Campus University, 00165 Rome, Italy

**Keywords:** endometriosis, adenogenesis, endometrium, adenomyosis, uterus

## Abstract

Endometriosis is a pathological condition characterized by the presence of the endometrial tissue, outside the uterine cavity. It affects nearly 10% of women of reproductive age and is responsible for infertility, chronic pain, and the weakening of the quality of life. Various pathogenetic mechanisms have been suggested; however, the essential pathogenesis of endometriosis remains insufficiently comprehended. A comprehensive literature search was conducted in databases such as PubMed, Scopus, and Web of Science up to December 2024. Inclusion criteria encompassed studies investigating the pathogenetic mechanisms of endometriosis, while exclusion criteria included reviews, case reports, and studies lacking primary data. The analyzed studies explored multiple pathogenetic mechanisms, including retrograde menstruation, coelomic metaplasia, embryological defects, stem cell involvement, and epigenetic modifications. Special emphasis was placed on the role of uterine adenogenesis factors in the development and progression of endometriosis. A deeper understanding of the various pathogenetic mechanisms underlying endometriosis is crucial for advancing targeted therapeutic strategies. Further research into uterine adenogenesis factors may provide new insights into the disease’s pathophysiology and pave the way for novel treatment approaches.

## 1. Introduction

Approximately 10–15% of women of reproductive age are affected by endometriosis. This condition can present symptoms such as intense pain, painful menstruation (dysmenorrhea), painful intercourse (dyspareunia), difficulty urinating (dysuria), infertility, dyschezia, and fatigue [1,2]. The exact cause remains unknown, but it may be linked to the complex development and maturation of the uterus. Regardless of its origin, there is broad consensus that endometriosis is linked to a localized inflammatory response, with vascularization at the site of invasion playing a critical role in its pathogenesis. While it is well established that endometriotic lesions consist of stromal and epithelial glandular cells, the development and specific characteristics of the cell types involved in the disease’s pathogenesis remain poorly understood [3].

In humans, the uterus is not fully developed or functionally differentiated at birth. While its basic structure is present, the uterus undergoes significant postnatal changes in size, morphology, and cellular organization. These developmental processes, influenced by hormonal cues during childhood, puberty, and later life, gradually prepare the uterus for its reproductive functions [4]. Endometriosis is often discovered during evaluations for infertility, but a direct causal relationship has not been definitively established. Laparoscopy, ideally with histological confirmation, remains the gold standard for diagnosis [5], which may lead to an overestimation of its prevalence and incidence among infertile women compared to the general population [6]. Women with otherwise “asymptomatic” endometriosis linked to underlying infertility causes may be disproportionately represented. Without attempts to conceive, these women might not have undergone laparoscopic diagnosis [7]. Moreover, high-quality clinical trials have shown no significant differences in spontaneous pregnancy rates among endometriosis patients receiving hormonal treatments [8]. This persistence may partly explain the lack of improvement in infertility. Additionally, the surgical removal of pelvic lesions offers only modest benefits for reproductive outcomes [9]. While current treatments for endometriosis primarily focus on symptom relief, such as pain management, surgical removal, and hormonal suppression, the development of a more targeted treatment, based on a deeper understanding of its pathological mechanisms, remains an urgent need.

The disease is estrogen-dependent, with the expression of estrogen receptor (ER)β significantly higher in EMS compared to normal endometrial tissues. Elevated levels of ERβ suppress ERα, leading to very high ERβ/ERα ratios [10]. ERβ plays a critical role in EMS pathogenesis by modulating apoptosis and the inflammasome. Furthermore, ERβ regulates mitochondrial function through nuclear respiratory factor-1 (NRF1) and helps control the basal expression of superoxide dismutase 2 (SOD2) to minimize oxidative stress [11]. ERβ also influences the expression of cyclooxygenase-2 (COX2), involved in inflammation, and matrix metalloproteinase 1 (MMP1), which contributes to tissue degradation, cell invasion, and growth [12,13]. Although endometriosis is not considered a tumor, many of its features, such as tissue invasion and neovascularization, are like those of malignant tumors. The microenvironment of endometriotic lesions is characterized by chronic inflammation, oxidative stress, and angiogenesis. These factors contribute to the pathogenesis of the disease and may explain the resistance to pharmacological treatments [14]. Given the complexities and gaps in understanding the pathogenesis of endometriosis, this review aims to consolidate and critically evaluate the foremost pathogenetic theories based on recent investigations. Specifically, we focus on the potential role of uterine adenogenesis factors in the development and progression of endometriosis. By shedding light on these mechanisms, we hope to contribute to a deeper understanding that could pave the way for more targeted and effective therapeutic strategies for this condition.

## 2. Methods

A comprehensive literature search was conducted using PubMed, Scopus, and Web of Science to identify relevant studies on the pathogenetic mechanisms of endometriosis. The search was performed up to December 2024. Only studies published in English were considered.

Inclusion criteria: Studies investigating the pathogenetic mechanisms of endometriosis, including theories related to retrograde menstruation, coelomic metaplasia, embryological defects, stem cell involvement, and epigenetic alterations. Original research articles involving human subjects or animal models provided that the latter offered translational insights applicable to human endometriosis. Studies providing molecular, histological, or clinical insights into the development and progression of endometriosis.

Exclusion criteria: Review articles, meta-analyses, case reports, and conference abstracts. Studies without primary data or those based solely on theoretical models. Animal studies lacking translational relevance to human endometriosis.

Selection Process: The selection process was performed in three stages:

Title and abstract screening to exclude irrelevant studies.

Full-text evaluation of potentially eligible articles.

Final selection of studies meeting all inclusion criteria.

## 3. Pathogenesis of Endometriosis: Mechanisms and Theories

Endometriosis is characterized by the presence of endometrial-like cells outside the uterus, leading to a chronic inflammatory response. It presents a wide range of clinical manifestations, with common symptoms including chronic abdominal or pelvic pain, painful menstruation (dysmenorrhea), painful intercourse (dyspareunia), and painful defecation (dyschezia) [1,14]. The specific symptoms can vary depending on the organ or site involved: abdominal wall pain, chest pain, hemoptysis, or bladder dysfunctions such as urgency, urinary frequency, and dysuria. In cases of intestinal endometriosis, patients may experience diarrhea, constipation, intestinal cramps, dyschezia, and, in rare cases, rectal bleeding. An increasing number of symptoms is associated with a higher probability of endometriosis diagnosis. In particular, dyschezia and dyspareunia are strongly predictive of deep infiltrating endometriosis (DIE) [15]. However, the clinical presentation is highly variable, ranging from asymptomatic forms to severe symptoms, contributing to the long delay between symptom onset and diagnosis, which can range from 7 to 12 years. One study found that 30% of patients were referred to a gynecologist at least twice before receiving a definitive diagnosis [16,17].

Research on endometriosis has gained significant attention in recent decades, but substantial challenges remain, particularly in achieving timely diagnosis. Factors such as diverse symptom presentation, the lack of standardized diagnostic criteria, and the underestimation of symptoms by both patients and healthcare providers contribute to prolonged diagnostic delays, averaging 4–10 years. These delays can exacerbate symptoms, reduce quality of life, and increase healthcare costs [6]. While historical and recent studies indicate slight improvements in diagnostic timelines, largely due to enhanced awareness and targeted initiatives such as advocacy campaigns, educational programs for clinicians, and public awareness efforts, significant work is still needed. Reducing diagnostic delays is critical for improving patient outcomes and the effective management of endometriosis [18]. Interestingly, recent data from our research group and from others have indicated some molecular markers that could be used as diagnostic tools for endometriosis [19].

The pathogenesis of endometriosis is multifactorial, involving genetic and cellular factors, as well as altered immunity (Table 1). The most widely accepted theory is Sampson’s retrograde menstruation theory, which suggests that endometrial cells flow through the fallopian tubes into the peritoneal cavity [20]. The retrograde menstruation theory proposes that during menstruation, some endometrial tissue flows backward through the fallopian tubes into the peritoneal cavity, causing endometriosis. This explains common sites of endometriosis, like the fallopian tubes and peritoneal wall, and is supported by the increased risk in women with uterine outflow obstruction and the frequent occurrence of blood in the pelvis during menstruation [2,21]. However, this theory fails to explain deep endometriosis, where lesions are deeper in the pelvic organs, or endometriosis in distant sites like the lungs and skin. It also does not apply to endometriosis in adolescents, newborns, women with Mayer–Rokitansky–Küster–Hauser syndrome, or men. Moreover, studies suggest that endometriotic tissue does not resemble an auto-transplant [22,23,24].

In 1942, Gruenwald proposed the theory of coelomic metaplasia, which suggests that the coelomic walls, such as the peritoneal serosa, are embryologically linked to the Müllerian ducts. According to this theory, endometriosis may develop in various coelomic derivatives due to a metaplastic process. Additionally, endocrine-disrupting chemicals might play a role in the transformation or activation of these coelomic cells, as suggested by several studies. This theory could help explain cases where retrograde menstrual flow is not possible [25,26].

The lymphatic and vascular spread theory suggests that endometrial cells could spread through lymphatic or blood vessels. In this model, it is proposed that menstrual endometrium, containing both epithelial and stromal cells, can enter these circulatory systems without disruption and then exit the vessels to implant in the muscular layers of organs. While deep endometriosis shares characteristics with cancer metastasis, such as tissue invasion, these are typical of tumor cells. However, there is currently no scientific evidence to support the notion that menstrual endometrium, originating from benign tissue, can perform such cancer-like functions [22].

The stem cell recruitment theory suggests that endometriosis may arise from stem cells, either from the uterine endometrium or the bone marrow. The first hypothesis proposes that stem cells from the uterine endometrium, specifically from the basal glands, are responsible for regenerating the endometrial epithelium and could contribute to endometriosis if misplaced. These epithelial stem cells are thought to reside in the basal layer at the endometrial/myometrial interface, although no specific markers have been identified for them [27]. The second hypothesis suggests that bone-marrow-derived stem cells are a key source of endometriosis, especially when these cells differentiate into both epithelial and stromal cells. If these stem cells are misplaced and end up in soft tissues instead of returning to the endometrium, endometriosis may develop. Experimental evidence supports this theory, with bone marrow stem cells also implicated in rare cases of endometriosis occurring outside the peritoneal cavity or in men [28,29].

The theory of embryonic cell remnants proposes that remnants of embryonic cells from the Wolffian or Müllerian ducts can develop into endometriotic lesions under the influence of estrogen. This theory suggests that abnormal differentiation or migration of the Müllerian ducts during embryogenesis could lead to the spread of primordial endometrial cells, which remain dormant until puberty, when estrogen activation causes their proliferation. This mechanism could explain the common sites of endometriosis, such as the deep peritoneum and the pouch of Douglas [30].

Recent studies have supported this theory, with findings of ectopic endometrial structures in female fetuses during autopsies. These structures were found in locations typically affected by endometriosis in adults, such as the Douglas pouch and rectovaginal septum. This provides strong evidence for the fetal origin of endometriosis [31,32,33].

Immunohistochemical studies of endometriotic structures have shown that these lesions have characteristics similar to ectopic endometrium, with both epithelial components (expressing estrogen receptor, CA-125, and cytokeratin 7) and stromal cells (positive for CD-10 and estrogen receptor). Interestingly, fetal endometrial cells from these patients showed identical staining patterns. These findings support the theory of embryonic cell remnants as a cause of endometriosis, showing that ectopic endometrial structures can persist outside the uterus during organogenesis [34]. Further supporting this theory, a study by de Jolinière et al. found ectopic endometrial glands in female fetuses, with similar immunohistochemical profiles [35]. In addition, previous studies have once again reported cases of fetal endometriosis, such as a cystic mass in the left ovary diagnosed in a fetus at 35 weeks [36]. The presence of displaced Müllerian tissue during organogenesis has been suggested by various research groups as a potential factor in the development of endometriosis. Additionally, a significant body of research has demonstrated a strong correlation between uterine malformations or Müllerian duct anomalies and the occurrence of endometriosis [37]. Moreover, genomic studies by our research group have shown that the transcriptome of endometriotic tissue expresses numerous genes related to embryogenesis in a manner distinct from normal endometrial tissue. Notably, this altered gene expression occurs independently of the hormonal cycle phase [38,39].

Based on these observations, it can be hypothesized that endometriosis results from disturbances in uterine embryogenesis during a critical period of morphogenesis. These disruptions could lead to the abnormal displacement of endometrial tissue outside the uterine cavity. However, the specific genetic and epigenetic factors responsible for these disturbances remain unidentified [40]. Estrogen, a key hormone in the development of the female reproductive system, may play a central role in this process. It is possible that altered estrogen levels, acting on a genetic predisposition, could trigger the formation of ectopic endometrial tissue during embryogenesis. Furthermore, there is substantial evidence linking exposure to endocrine disruptors, substances that mimic estrogen, to both uterine abnormalities and endometriosis [41,42,43].

**Table 1 ijms-26-02076-t001:** Major theories on the origin of endometriosis.

Theory	Proposed Mechanism	References
Retrograde menstruation	Retrograde menstrual flow allows the implantation of endometrial tissue in the peritoneal cavity	[19]
Coelomic metaplasia	Structures derived from the coelomic epithelium may undergo transformation into endometriotic tissue via metaplasia	[25]
Vascular dissemination	Endometrial cells spread through blood or lymphatic vessels, colonizing distant anatomical sites	[21]
Stem cell involvement	Endometrial or hematopoietic stem cells may differentiate into endometriotic lesions in ectopic locations	[28,29]
Embryonic remnants	Residual embryonic cells from Wolffian or Müllerian ducts may develop into endometriotic lesions under estrogen stimulation	[31,32,33,34,35]

## 4. Development of the Female Reproductive System

The development of the human female reproductive system is a topic of significant academic and clinical importance. Congenital malformations of this anatomical region often arise from disruptions in normal morphogenetic processes and the molecular mechanisms that support them. Even though relatively rare, many of these congenital anomalies are associated with defects in Müllerian duct (MD) development. Some occur spontaneously, while others are induced by exposure to endocrine-disrupting substances, particularly those with estrogenic activity [44].

Most of the female reproductive tract develops from the Müllerian ducts (MD), which arise as coelomic epithelial invaginations on the urogenital ridges around 5–6 weeks of gestation [45] (Figure 1). These invaginations form the ostia of the uterine tubes. The MDs grow caudally, using the Wolffian ducts (WD) as “guide wires”, as the WDs are essential for the caudal migration of the MDs toward the urogenital sinus (UGS). During migration, the MDs and WDs are initially separated by the mesenchyme of the urogenital ridge. However, further caudally, their epithelia come closer together, eventually sharing a common basement membrane. At the tip of the MDs, direct contact is established with the WD epithelium. The fusion of the MDs with the UGS, critical for forming the uterovaginal canal, occurs by the 8th week of gestation. Failures during this stage, such as unsuccessful fusion, can lead to malformations like vaginal agenesis [46]. During the fusion process, a temporary epithelial septum separates the MD lumina, but this septum disappears by the 9th week, forming a single uterovaginal canal lined by undifferentiated columnar epithelium of Müllerian origin. Mutations in key genes such as Lhx1 and Lhfpl2 can disrupt this process, leading to degeneration of the ducts or malformations of the reproductive tract [47,48].

The uterine tubes, also known as Fallopian tubes, consist of four main sections: the infundibulum, funnel-shaped with fimbriae; the ampulla, where fertilization typically occurs; the isthmus, located near the uterus; and the intramural portion, which traverses the uterine wall and ends at the uterotubal junction. These structures develop from the cranial portions of the MD, while the caudal segments fuse to form the uterovaginal canal. The fimbriae of the tubes arise from the irregular openings of the MD into the abdominal cavity [49,50,51].

During development, the fetal tubal epithelium initially consists of a single layer of columnar cells. Over time, mucosal folds develop and become especially prominent in infundibulum and ampulla by the fourteenth week of gestation. Concurrently, the surrounding mesenchyme differentiates into an inner stromal layer and an outer layer of circularly arranged smooth muscle [52]. In adulthood, the tubal epithelium includes ciliated cells, secretory cells, and intercalated (“peg”) cells. During fetal life, epithelium expresses differentiation markers such as keratins 7, 8, and 19, starting from the ninth week. Androgen receptors and ESR1 appear around the fourteenth week, whereas the progesterone receptor is undetectable between the eighth and twenty-first weeks, although it can be artificially induced under experimental conditions [53].

Numerous cellular and molecular mechanisms are involved in the development and differentiation of the MD in the female reproductive organs. Despite significant progress in recent years in understanding the signaling pathways that regulate the development of the female reproductive system, many areas remain poorly understood, particularly regarding the specific processes that coordinate the formation and maturation of reproductive organs [54].

Several signaling pathways have been identified as playing a crucial role in the development of the MDs and uterine tubes, including bone morphogenetic proteins (BMPs), the WNT signaling pathway, transforming growth factor-β (TGF-β), the PI3K/Akt pathway, G-protein-coupled receptors, and fibroblast growth factors (FGFs). These molecular mechanisms are involved in various stages of development, from cell proliferation to differentiation and the formation of the final anatomical structures. In particular, BMP signaling is essential for the proper development and function of the female reproductive system [55]. The BMP pathway regulates several critical aspects, including the formation of the MDs, their migration and fusion, and the subsequent differentiation of reproductive organs. Additionally, BMP signaling interacts with other pathways, such as WNT, to maintain the homeostasis of the reproductive system throughout both embryonic and postnatal development. While much has been learned about these signaling pathways, a complete understanding of how they interact in a coordinated manner remains an area of intense research [56,57,58].

The WNT protein family plays a critical role in numerous developmental processes and in tissue homeostasis. Among these, WNT4 plays a pivotal role in shaping the female phenotype during fetal development and in sustaining the integrity of Müllerian and reproductive tissues [59]. Alterations or the dysregulation of WNT4 are associated with conditions such as sex reversal syndromes, highlighting its crucial function in female sex determination. Furthermore, WNT4 is involved in various gynecological pathologies, including uterine fibroids, endometriosis, infertility, and tumors [60]. WNT4 also plays a significant role in processes such as decidualization, implantation, and gestation, which are essential for normal reproduction. However, abnormal activation of WNT4 signaling has been linked to the development of gynecological and breast tumors. Single nucleotide polymorphisms (SNPs) in the WNT4 gene have been strongly associated with these conditions and may represent a point of connection between estrogen signaling and WNT4-mediated pathways, contributing to the upregulation of WNT4 activity. SNPs in the 1p36 region of the human genome, which includes the WNT4 gene, are linked to an increased risk of various gynecological conditions. Genome-wide association studies (GWASs) have identified SNPs in this region as associated with endometriosis (often accompanied by infertility), uterine fibroids, ovarian cancer, and pelvic organ prolapse [61]. Notably, these variants increase the risk for endometriosis and ovarian cancer but appear to have a protective effect against pelvic organ prolapse and are associated with slightly longer gestation durations (approximately two additional days per allele) [62,63,64]. The primary SNP in most studies is located within the WNT4 gene locus, and its associations have been confirmed through multiple independent studies and meta-analyses. These findings are consistent across diverse ethnic and geographic populations, underscoring the robust nature of the link between SNPs in this genomic region and gynecological health outcomes.

In each developmental or pathological context, the expression and activity of WNT4 signaling are regulated by tissue-specific pathways [65].

## 5. Uterine Adenogenesis

Uterine adenogenesis, the development of endometrial glands, is a critical process for the formation and functionality of the adult uterus [66,67]. In humans, this process begins during fetal life, continues postnatally, and completes at puberty. In other species, such as rodents, sheep, and pigs, adenogenesis primarily occurs postnatally. Endometrial gland formation involves the differentiation and budding of glandular epithelium from the luminal epithelium, followed by invagination, tubular morphogenesis, and branching into the uterine stroma. This process is regulated by both local and systemic factors, including site-specific changes in cell proliferation, extracellular matrix (ECM) remodeling, and paracrine signaling between cells and the ECM [68]. Several hormones are involved in regulating adenogenesis, including prolactin and estradiol-17β, which influence cell proliferation, extracellular matrix remodeling, and paracrine signaling. These hormones play a significant role in regulating both the development and function of endometrial glands in mammals, with receptors for these hormones being crucial for the process. The interaction between these hormones and specific molecular pathways ensures the proper formation of the glands, which are essential for successful implantation and pregnancy [69]. Uterine gland secretions play a fundamental role in supporting blastocyst implantation by providing essential nutrients and signaling molecules necessary for early embryonic development [70,71]. While some evidence suggests that uterine glands may contribute to the decidualization process by modulating the endometrial environment, their direct role remains incompletely understood. Further studies are needed to elucidate the precise mechanisms through which uterine gland secretions influence decidualization. In mice, leukemia inhibitory factor (LIF), produced exclusively by uterine glands on day 4 of pregnancy, is critical for blastocyst adhesion to the uterine luminal epithelium (LE) [72,73]. Its absence, as observed in Lif knockout mice, results in infertility due to the failure of embryonic implantation [74,75]. Similarly, in neonatal progesterone-treated mice, which develop aglandular uteri (Progesterone-treated Uterine Gland Knockout: PUGKO), blastocysts hatch but fail to adhere to the LE, and no decidualization is observed, even after artificial stimulation or intrauterine treatment with LIF [76].

During pregnancy, species such as sheep and pigs exhibit hyperplasia and hypertrophy of endometrial glands, necessary for producing histotroph, a critical nutrient source for the conceptus. Alterations or defects in endometrial gland morphogenesis, caused by genetic anomalies, epigenetic influences, or endocrine disorders, can impair uterine functionality and contribute to unexplained peri-implantation embryonic losses in humans and animals. Understanding the mechanisms that regulate uterine adenogenesis is, therefore, essential for improving reproductive health and addressing infertility and reproductive failures [66,69].

Adenogenesis involves the coordinated expression of several critical genes, including forkhead box A2 (Foxa2), Wnt4, Wnt5a, Wnt7a, and E-cadherin (Cdh1), among others. One of the most well-studied genes in this process is Foxa2, a transcription factor specifically expressed in uterine glands. The role of FOXA2 in uterine gland development is influenced by the timing of its deletion. The conditional deletion of FOXA2 in the uterus using Pgr-Cre or Wnt7a-Cre results in a significant reduction or complete absence of uterine glands, demonstrating its critical role in glandular development and uterine function [77]. However, studies utilizing Ltf-Cre mice have shown that FOXA2 deletion at later stages does not entirely abolish gland formation, suggesting that the impact of FOXA2 on uterine adenogenesis may be dependent on the timing and specificity of gene deletion [78,79]. The Wnt signaling pathway, particularly Wnt4, Wnt5a, and Wnt7a, is also fundamental for adenogenesis. Studies have shown that the deletion of Wnt7a or Wnt4 in the mouse uterus results in either a complete absence or a significant reduction in the number of uterine glands [80,81]. Additionally, E-cadherin (Cdh1), a key molecule involved in cell-cell adhesion, is essential for proper gland development. Conditional ablation of Cdh1 results in the loss of uterine glands in neonatal mice, highlighting the importance of cell adhesion molecules in maintaining the integrity and functionality of the developing uterine glands [82].

## 6. Expression Patterns of Different Adenogenesis Factors in Endometriosis and Normal Endometrium

Our previous studies investigated stromal and epithelial samples from deep endometriosis lesions, along with normal endometrial epithelium and stroma used as controls. The analysis revealed no significant morphological differences between samples taken during the proliferative and secretory phases. Endometriotic lesions consistently showed the presence of both endometriotic glands and stroma, regardless of the lesion site. The glands displayed a classical endometrioid appearance, while the stroma closely resembled eutopic endometrial stroma in inactive or proliferative states.

The immunohistochemical analysis focused on the expression of IFN-τ, FGF-7, FGF-10, FGF-23, and HGF in both normal endometrial tissues and deep endometriosis samples. Specific cytoplasmic immunopositivity was detected for all antibodies tested. The findings demonstrated that FGF-7, FGF-10, and HGF exhibited significantly higher expression levels in the epithelium and stroma of normal controls compared to endometriosis samples. In contrast, FGF-23 and IFN-τ showed significantly higher expression in the ectopic endometrial stroma of endometriosis samples relative to eutopic endometrium, whereas no notable differences in epithelial expression were observed between the two tissue types.

The statistical analysis of the normal endometrium during the proliferative and secretory phases revealed no significant differences in the expression of these growth factors within either the epithelial or stromal compartments. Similarly, this pattern was observed in endometriotic tissues, suggesting that fluctuations in estrogen and progesterone levels do not substantially influence the expression of these factors in normal or endometriotic samples, highlighting molecular distinctions between deep endometriosis and normal endometrium. These observations support the notion that endometriotic tissues, including both epithelial and stromal components, exhibit a distinct phenotype compared to the eutopic endometrium. This reinforces the hypothesis that alterations in the molecular mechanisms regulating adenogenesis and the survival of endometrial structures are closely linked to the development and persistence of endometriotic lesions outside the uterus [83]. Further evidence suggests that FGF, IFN-τ, and HGF not only regulate uterine adenogenesis but also contribute to the establishment and persistence of endometriotic lesions. Mechanistic studies indicate that FGFs play a crucial role in epithelialmesenchymal interactions, which are dysregulated in endometriosis. In particular, FGF-7 and FGF-10 promote epithelial proliferation and survival, while aberrant signaling may contribute to the invasive potential of endometriotic lesions [84,85]. Similarly, HGF has been implicated in the modulation of stromal-epithelial communication, and its overexpression in ectopic endometrial tissue is associated with increased cell motility and invasiveness [86,87]. IFN-τ, originally studied for its role in pregnancy recognition in ruminants, has been reported to have immunomodulatory effects that could influence the inflammatory microenvironment of endometriotic lesions [88]. The interplay between these factors may facilitate the survival of ectopic endometrial tissue and contribute to the chronic inflammatory state observed in endometriosis.

In a preceding study, we analyzed stromal and epithelial samples from endometriosis lesions and normal endometrium controls during both the proliferative and secretory phases. Immunohistochemistry was used to assess PRL-R and GH hormones, as well as IGF1 and IGF2 expressions, in normal endometrium and endometriosis tissues. IGF1 and IGF2 showed significantly higher expression in both epithelium and stroma of controls compared to endometriosis samples. For PRL-R, this pattern was observed only in the epithelium, while GH displayed significantly higher expression in the epithelium of endometriosis samples compared to controls. The statistical analysis further revealed that in normal uterine tissue, GH and PRL-R expression in the epithelium was significantly higher during the secretory phase compared to the proliferative phase. However, this phase-dependent variation was absent in endometriotic tissues. The correlation data generated provide insights into some of the molecular mechanisms involved in the adenogenesis and survival of endometriotic structures outside the uterus, shedding light on the processes that sustain the development and persistence of these lesions [89].

Finally, we analyzed and compared the expression of various proteoglycans and specific glycosaminoglycans (GAGs), key components of the extracellular matrix, between deep endometriotic lesions and normal endometrial tissue. These GAGs play a crucial role in the interaction between the epithelium and stroma, a process essential for proper uterine gland morphogenesis. The study examined the expression of CSPG4 (chondroitin sulfate proteoglycan 4), CS-56 (a chondroitin sulfate), HEP (heparan sulfate), keratan sulfate, and hyaluronic acid in endometriosis-affected tissues compared to normal endometrium. CSPG4 and CS-56 showed significantly higher expression in the epithelium of endometriosis samples, while HEP was more highly expressed in the epithelium of normal samples. Stromal expression for these markers showed no significant differences. Keratan sulfate was largely absent in the epithelium of endometriosis samples, with stromal expression predominantly low compared to the medium or high intensity observed in normal endometrium. Hyaluronic acid expression was consistently lower in both the epithelium and stroma of endometriosis samples. Logistic regression analysis demonstrated that high epithelial expression of CSPG4, CS-56, and HEP increased the likelihood of association with endometriosis, while high expression of keratan sulfate and hyaluronic acid in both the epithelium and stroma correlated with a reduced likelihood of belonging to the endometriosis group.

Although HEP showed higher epithelial expression in normal endometrium, the logistic regression analysis identified that relatively higher expression levels of CSPG4, CS-56, and HEP in the epithelium were predictive of endometriosis, while higher keratan sulfate and hyaluronic acid expression in both epithelium and stroma were associated with a reduced likelihood of endometriosis. All the data described in this section are summarized in Table 2.

These findings highlight significant alterations in marker expression, providing insights into endometriosis pathophysiology and suggesting potential diagnostic markers [90].

## 7. Conclusions

The pathogenesis of endometriosis is complicated and comprises many causes and processes, which can occur simultaneously. Many theories have been investigated, but there has been no individual theory up to now, which could justify all aspects of endometriosis. Endometriosis growth and progression probably incorporate elements from all the recorded pathogenetic theories. In this context, the role of uterine adenogenetic factors, also based on the data produced about their differential expression in respect to endometriosis tissue, is a research area to be deeply investigated. An improved comprehension of the pathogenesis of this cryptic but, unfortunately, very common reproductive disorder can lead to a better treatment and an enhanced quality of life for patients experiencing endometriosis.

## Figures and Tables

**Figure 1 ijms-26-02076-f001:**

Uterine formation.

**Table 2 ijms-26-02076-t002:** Immunohistochemical comparison between eutopic and ectopic endometrium.

Marker	Eutopic Epithelium/Stroma	Ectopic Epithelium/Stroma
FGF7	High/High	Low/Low
FGF10	High/High	Low/Low
FGF23	High/Low	High/High
IFN-τ	High/Low	High/High
HGF	High/High	Low/Low
PRL-R	High/Low	Low/Low
GH	Low/Low	High/Low
IGF1	High/High	Low/Low
IGF2	High/High	Low/Low
CSPG4	High/Low	High/Low
CS-56	High/Low	High/Low
HEP	High/Low	High/Low
Keratan	Low/Low	Low/Low
Hyaluronic acid	Low/Low	Low/Low

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
