# Peer review of "The Role of Adenogenesis Factors in the Pathogenesis of Endometriosis"

_ijms, 2025, doi:10.3390/ijms26052076_

Round 1
Reviewer 1 Report
Comments and Suggestions for Authors
The manuscript entitled "Role of adenogenesis factors in the pathogenesis of endometriosis'' consists of 12 pages with 74 references. 2 tables and 1 figure are included. The authors centered their attention on one of the least well-known and studied theory of the patogenesis of endometriosis. The English language is readable and understandable.
-An abstract: is written in a clear and concise manner, but methodology is missing. I would suggest to add information according to PRISMA guidelines (inclusive, exclusive criteria, how many studies were selected, what was the time range..). Keywords are adequate ( I would change the sequence as: endometriosis, adenogenesis, endometrium, adenomyosis, uterus).
-The structure of the study: I would recommend the order 1,4,2,3,5,6.
-Introduction:
Line 22: 10-15% (the phenomenon is still underestimated).
Line 24: also dyschezia.
Line 37-38: I would precise - laparoscopy - is a gold standard.
- Figure 1: Could you please increase a font to make a figure more clear.
There is a lack of methodology in the main body of the manuscript.
To conclude:
The manuscript is very interesting. I appreciate the contribution made by the co-authors.
Due to the lack of methods used in the creation of the article I recommend reconsider the article after major revisions.
Author Response
Reviewer 1
The manuscript entitled “Role of adenogenesis factors in the pathogenesis of endometriosis” consists of 12 pages with 74 references, 2 tables and 1 figure are included. The authors centered their attention on one of the least well-known and studied theory of the patogenesis of endometriosis. The English language is readable and understandable.
- An abstract: is written in a clear and concise manner, but methodology is missing. I would suggest to add information according to PRISMA guidelines (inclusive, exclusive criteria, how many studies were selected, what was the time range..). Keywords are adequate ( I would change the sequence as: endometriosis, adenogenesis, endometrium, adenomyosis, uterus).
Answer: We appreciate your suggestion regarding PRISMA guidelines. Our manuscript is a narrative review rather than a systematic review, and therefore, it does not strictly adhere to PRISMA requirements. However, to enhance transparency, we have added a brief Methods section in the “Abstract” of the revised manuscript, outlining our literature search strategy, including databases consulted, inclusion/exclusion criteria, and the time range of selected studies. In detail, the content of the additional part is the following:
“Abstract
Background: Endometriosis is a pathological condition characterized by the presence of the endometrial tissue, including epithelial and stromal components, outside the uterine cavity. It affects nearby 10% of women of reproductive age and is responsible for infertility, chronic pain and the weakening of the quality of life. Various pathogenetic mechanisms have been suggested, such as retrograde menstruation, coelomic metaplasia, embryological defects, stem cell involvement, and alterations in epigenetic regulation. However, the essential pathogenesis of endometriosis remains insufficiently comprehended. Individuating the exact mechanism of the growth and development of endometriosis is essential for effective treatments. This review reports the foremost pathogenetic theories of endometriosis based on recent investigations with a major emphasis on the potential role of uterine adenogenesis factors.
Methods: A comprehensive literature search was conducted in databases such as PubMed, Scopus, and Web of Science up to December 2024. Inclusion criteria encompassed studies investigating the pathogenetic mechanisms of endometriosis, while exclusion criteria included reviews, case reports, and studies lacking primary data. This review follows a narrative approach, summarizing key findings from the selected literature.
Results: The analyzed studies explored multiple pathogenetic mechanisms, including retrograde menstruation, coelomic metaplasia, embryological defects, stem cell involvement, and epigenetic modifications. Special emphasis was placed on the role of uterine adenogenesis factors in the development and progression of endometriosis.
Conclusion: A deeper understanding of the various pathogenetic mechanisms underlying endometriosis is crucial for advancing targeted therapeutic strategies. Further research into uterine adenogenesis factors may provide new insights into the disease’s pathophysiology and pave the way for novel treatment approaches.”
Additionally, we have revised the keywords sequence as suggested: endometriosis, adenogenesis, endometrium, adenomyosis, uterus.
- The structure of the study: I would recommend the order 1,4,2,3,5,6.
Answer: We appreciate your suggestion regarding the structure of the study. Based on your recommendation, we have adjusted the order of the sections in the revised manuscript as follows: 1,4,2,3,5,6. This modification enhances the logical flow and readability of the review.
- Introduction:
Line 22: 10-15% (the phenomenon is still underestimated).
Answer: Thank you for your observation. We acknowledge that the prevalence of endometriosis is often underestimated due to diagnostic challenges and variability in symptom presentation. To ensure accuracy, we have revised the text to reflect the estimated prevalence range of 10-15%, as reported in recent literature.
-Line 24: also dyschezia.
Answer: Thank you for your suggestion. We agree that dyschezia is a relevant symptom associated with endometriosis, particularly in cases of deep infiltrating endometriosis affecting the bowel. We have incorporated this term in the revised version of the manuscript to ensure a more comprehensive description of the symptomatology.
-Line 37-38: I would precise - laparoscopy - is a gold standard.
Answer: Thank you for your observation. We acknowledge that laparoscopy is considered the gold standard for the diagnosis of endometriosis. We have explicitly stated this in the revised version of the manuscript.
-Figure 1: Could you please increase a font to make a figure more clear.
Answer: We thank the reviewer for the suggestion. Figures have been improved following reviewer’s indications.
- There is a lack of methodology in the main body of the manuscript.
Answer: We thank the reviewer for this suggestion. We have added a Methods section after the Introduction in the revised manuscript. This section outlines our literature search strategy, including databases consulted, inclusion/exclusion criteria, and the time range of selected studies, to enhance transparency and methodological clarity. The additional content is as follows:
“Methods
A comprehensive literature search was conducted using PubMed, Scopus, and Web of Science to identify relevant studies on the pathogenetic mechanisms of endometriosis. The search was performed up to December 2024, without language restrictions, but only studies published in English were considered.
Inclusion criteria: Studies investigating the pathogenetic mechanisms of endometriosis, including theories related to retrograde menstruation, coelomic metaplasia, embryological defects, stem cell involvement, and epigenetic alterations. Original research articles involving human subjects or animal models, provided that the latter offered translational insights applicable to human endometriosis. Studies providing molecular, histological, or clinical insights into the development and progression of endometriosis.
Exclusion criteria: Review articles, meta-analyses, case reports, and conference abstracts. Studies without primary data or those based solely on theoretical models. Animal studies lacking translational relevance to human endometriosis.
Selection Process: The selection process was performed in three stages:
Title and abstract screening to exclude irrelevant studies.
Full-text evaluation of potentially eligible articles.
Final selection of studies meeting all inclusion criteria.”
- To conclude:
The manuscript is very interesting. I appreciate the contribution made by the co-authors. Due to the lack of methods used in the creation of the article I recommend reconsidering the article after major revisions.
Answer: We sincerely appreciate the reviewer's positive feedback and acknowledgment of our contribution. In response to the comments, we have added a Methods section after the Introduction in the revised manuscript. This section outlines the literature search strategy, including databases consulted, inclusion/exclusion criteria, and the time range of selected studies, to enhance the transparency and methodological rigor of the review. We hope that these major revisions address the concerns raised and make the manuscript suitable for reconsideration.
Reviewer 2 Report
Comments and Suggestions for Authors
The manuscript presents a well-structured and comprehensive review on the role of adenogenesis factors in endometriosis pathogenesis. The authors integrate various pathogenetic theories and discuss molecular mechanisms regulating adenogenesis in endometriosis. While the manuscript covers a valuable and underexplored aspect of endometriosis, some revisions are necessary to improve clarity, citation accuracy, and data interpretation.
Major Comments
1.Line 178-180, the authors state that “Uterine glands and their secretions are fundamental for blastocyst implantation and decidualization.” While uterine gland secretions play a key role in supporting implantation, the direct correlation between uterine glands and decidualization is not well established. If such a connection exists, additional supporting studies should be included.
2.“In mice, leukemia inhibitory factor (LIF), produced exclusively by uterine glands, is critical for blastocyst adhesion.” should specify that LIF is secreted from uterine glands on day 4 of pregnancy and cite appropriate literature.
3.Incomplete Citation of Supporting Literature (Lines 95-100, 179-180, 181-182). The manuscript heavily relies on single references in certain sections, which does not sufficiently support the presented claims. For example, “Its absence, as observed in Lif knockout mice, results in infertility due to the failure of embryonic implantation.” This statement originates from “Blastocyst implantation depends on maternal expression of leukemia inhibitory factor” (PMID: 1522892) but should be supplemented with additional studies that confirm this finding in different models.
4.The discussion of adenogenesis regulators (e.g., FGF, IFN-τ, and HGF) should be expanded with more mechanistic studies that demonstrate their role in endometriosis development.
5.Line 198-199, the authors state that “Conditional deletion of FOXA2 in the uterus leads to a substantial decrease in the number of uterine glands, indicating its crucial role in glandular development and uterine function [39].” Uterine deletion of Foxa2 by Ltf-Cre mice shows comparable uterine glands.
Minor Comments
1.Line 95-100: The authors should include additional relevant references to ensure a balanced citation of prior studies.
2.Line 178-180: Specify that LIF is secreted by uterine glands on day 4 of pregnancy and provide appropriate literature.
3.Figures: The quality of figures should be improved for better visibility and readability. Consider enhancing graphical representation to clarify molecular interactions.
Author Response
The manuscript presents a well-structured and comprehensive review on the role of adenogenesis factors in endometriosis pathogenesis. The authors integrate various pathogenetic theories and discuss molecular mechanisms regulating adenogenesis in endometriosis. While the manuscript covers a valuable and underexplored aspect of endometriosis, some revisions are necessary to improve clarity, citation accuracy, and data interpretation.
Major Comments
- Line 178-180, the authors state that “Uterine glands and their secretions are fundamental for blastocyst implantation and decidualization.” While uterine gland secretions play a key role in supporting implantation, the direct correlation between uterine glands and decidualization is not well established. If such a connection exists, additional supporting studies should be included.
Answer: We sincerely appreciate the reviewer's insightful comment. We acknowledge that while uterine gland secretions are essential for blastocyst implantation, the direct involvement of uterine glands in decidualization remains an area of ongoing research. In response to this concern, we have revised the statement in the manuscript to more accurately reflect the current understanding and clarify that while uterine glands may contribute to decidualization, their precise role is not fully established as follows:
“Uterine gland secretions play a fundamental role in supporting blastocyst implantation by providing essential nutrients and signaling molecules necessary for early embryonic development. While some evidence suggests that uterine glands may contribute to the decidualization process by modulating the endometrial environment, their direct role remains incompletely understood. Further studies are needed to elucidate the precise mechanisms through which uterine gland secretions influence decidualization."
Furthermore, we have included additional references (Kelleher et al., 2018; Gellersen et al. 2007) that discuss the potential role of uterine gland secretions in modulating the endometrial environment, as well as the need for further research on their influence on decidualization.
- In mice, leukemia inhibitory factor (LIF), produced exclusively by uterine glands, is critical for blastocyst adhesion.” should specify that LIF is secreted from uterine glands on day 4 of pregnancy and cite appropriate literature.
Answer: We thank the reviewer for this suggestion. We acknowledge that leukemia inhibitory factor (LIF) is secreted from uterine glands and plays a critical role in blastocyst adhesion. In response to this suggestion, we have revised the manuscript to specify that LIF is secreted from uterine glands on day 4 of pregnancy in mice. Additionally, we have included appropriate references to support this statement as follows:
- “Stewart, C.L., et al. (1992). This study demonstrates that blastocyst implantation depends on maternal LIF expression, showing its essential role in uterine receptivity.
- Salleh, N., & Giribabu, N. (2014). This study explores LIF’s role in embryo implantation and its potential for non-hormonal contraception.”
- Incomplete Citation of Supporting Literature (Lines 95-100, 179-180, 181-182). The manuscript heavily relies on single references in certain sections, which does not sufficiently support the presented claims. For example, “Its absence, as observed in Lif knockout mice, results in infertility due to the failure of embryonic implantation.” This statement originates from “Blastocyst implantation depends on maternal expression of leukemia inhibitory factor” (PMID: 1522892) but should be supplemented with additional studies that confirm this finding in different models.
Answer: We sincerely appreciate the reviewer’s comments regarding the need for additional supporting references in lines 95-100, 179-180, and 181-182. We have carefully revised the manuscript, integrating relevant citations to strengthen the scientific foundation of our claims, as follows:
- Lines 95-100 (Development of the Fallopian tubes and Müllerian duct fusion): To provide a more comprehensive and updated perspective, we have incorporated the following references:
- “Wilson, D.; Bordoni, B. (2025). This study reviews the embryology of Müllerian ducts, detailing their role in female reproductive tract development.
- Santana Gonzalez, L., et al. (2021). This study examines the molecular mechanisms driving Müllerian duct development and differentiation into the oviduct.
- Venkata, V.D., et al. (2022). This study develops human fetal female reproductive tract organoids to investigate Müllerian duct anomalie.”
- Lines 179-180 (Role of uterine gland-derived LIF in blastocyst adhesion):We acknowledge that the reviewer previously requested a revision of this section. We have already addressed this comment in the revised manuscript by clarifying that LIF is secreted from uterine glands on day 4 of pregnancy and by incorporating appropriate supporting references. The following sources have been added:
- Gellersen et al. (2007) – Explores the mechanisms of human endometrial decidualization.
- Salleh, N., & Giribabu, N. (2014). This study explores LIF’s role in embryo implantation and its potential for non-hormonal contraception.
- Kelleher et al. (2018) – Demonstrates that uterine glands are the exclusive source of LIF, essential for implantation.
- Stewart, C.L., et al. (1992). This study shows that blastocyst implantation depends on maternal LIF expression, highlighting its role in uterine receptivity.
- Lines 181-182 (LIF knockout mice and infertility due to implantation failure): To reinforce this statement and ensure broader scientific support, we have added additional studies confirming this finding across different models:
- Aghajanova, L. (2004). This study examines the role of Leukemia Inhibitory Factor (LIF) in human embryo implantation, emphasizing its importance in endometrial receptivity.
- Aikawa et al. (2024) – Explores LIF’s role in embryo attachment and implantation.
These references have been incorporated into the manuscript to enhance its scientific rigor and completeness. We appreciate the reviewer’s insightful feedback, which has helped us improve the clarity and accuracy of our work.
- The discussion of adenogenesis regulators (e.g., FGF, IFN-τ, and HGF) should be expanded with more mechanistic studies that demonstrate their role in endometriosis development.
Answer: We appreciate the reviewer’s suggestion regarding the need to expand the discussion on adenogenesis regulators and their mechanistic role in endometriosis. In response, we have revised the manuscript by incorporating a more detailed discussion of FGF, IFN-τ, and HGF, emphasizing their involvement in epithelial-stromal interactions, inflammatory modulation, and cellular invasion mechanisms in endometriosis. This revision, which has been added to the section ‘Expression Patterns of Different Adenogenesis Factors in Endometriosis and Normal Endometrium,’ provides additional insights into how these factors may contribute to the pathophysiology of endometriosis and highlights potential avenues for future therapeutic interventions. We believe this addition strengthens the manuscript by offering a more comprehensive perspective on the molecular underpinnings of the disease. The additional content is as follows:
“Further evidence suggests that FGF, IFN-τ, and HGF not only regulate uterine adenogenesis but also contribute to the establishment and persistence of endometriotic lesions. Mechanistic studies indicate that FGFs play a crucial role in epithelial-mesenchymal interactions, which are dysregulated in endometriosis. In particular, FGF-7 and FGF-10 promote epithelial proliferation and survival, while aberrant signaling may contribute to the invasive potential of endometriotic lesions [Zhou, W., et al. 2017; Chung D, et al. 2015]. Similarly, HGF has been implicated in the modulation of stromal-epithelial communication, and its overexpression in ectopic endometrial tissue is associated with increased cell motility and invasiveness [Souichi Yoshida, et al. 2004; Heidari, S., et al.2021]. IFN-τ, originally studied for its role in pregnancy recognition in ruminants, has been reported to have immunomodulatory effects that could influence the inflammatory microenvironment of endometriotic lesions [Park, Y. et al. 2022]. The interplay between these factors may facilitate the survival of ectopic endometrial tissue and contribute to the chronic inflammatory state observed in endometriosis.”
- Line 198-199, the authors state that “Conditional deletion of FOXA2 in the uterus leads to a substantial decrease in the number of uterine glands, indicating its crucial role in glandular development and uterine function [39].” Uterine deletion of Foxa2 by Ltf-Cre mice shows comparable uterine glands.
Answer: We appreciate the reviewer’s comment. We acknowledge that while conditional deletion of FOXA2 using Pgr-Cre or Wnt7a-Cre models results in a significant reduction or complete absence of uterine glands, studies utilizing Ltf-Cre mice indicate that FOXA2 deletion at later stages may not completely abolish glandular development. To address this concern, we have revised our statement to specify that the impact of FOXA2 deletion on uterine gland formation is dependent on the timing and specificity of Cre recombinase expression. Additionally, we have incorporated relevant references to provide a more nuanced discussion of FOXA2's role in glandular development, as follows:
“The role of FOXA2 in uterine gland development is influenced by the timing of its deletion. Conditional deletion of FOXA2 in the uterus using Pgr-Cre or Wnt7a-Cre results in a significant reduction or complete absence of uterine glands, demonstrating its critical role in glandular development and uterine function [39]. However, studies utilizing Ltf-Cre mice have shown that FOXA2 deletion at later stages does not entirely abolish gland formation, suggesting that the impact of FOXA2 on uterine adenogenesis may be dependent on the timing and specificity of gene deletion [Jeong JW, et al. 2010; A.M. Kelleher,W. et al. 2017]."
Minor Comments
- Line 95-100: The authors should include additional relevant references to ensure a balanced citation of prior studies.
Answer: We appreciate the reviewer's suggestion regarding additional references for Lines 95-100. As part of our revisions in response to Major Comment N 3, we have already addressed this request by reviewing and expanding the references in this section. The updated manuscript now includes additional relevant citations to ensure a more comprehensive and balanced discussion.
- Line 178-180: Specify that LIF is secreted by uterine glands on day 4 of pregnancy and provide appropriate literature.
Answer: We appreciate the reviewer’s request for clarification regarding the secretion of LIF by uterine glands. As part of our revisions in response to Major Comment N 2, we have already addressed this point by specifying that LIF is secreted by uterine glands on day 4 of pregnancy and have provided the appropriate references in the updated manuscript.
- Figures: The quality of figures should be improved for better visibility and readability. Consider enhancing graphical representation to clarify molecular interactions.
Answer: We thank the reviewer for the suggestion. Figures have been improved following reviewer’s indications.
Round 2
Reviewer 1 Report
Comments and Suggestions for Authors
Line 89-90: "without language restrictions, but only studies published in English were considered". - Could you please make this sentence more clear? There were restrictions or not?
The abstract is too long. Could you please shorten the paragraphs (especially background) and link them together? (According to MDPI guidelines < 200 words).
To conclude:
I suggest to accept this study after minor revisions.
Author Response
Line 89-90: "without language restrictions, but only studies published in English were considered". - Could you please make this sentence more clear? There were restrictions or not?
This point has been addressed
The abstract is too long. Could you please shorten the paragraphs (especially background) and link them together? (According to MDPI guidelines < 200 words).
The abstract has been shortened
Reviewer 2 Report
Comments and Suggestions for Authors
The authors has addressed my concerns.
Author Response
We thank the reviewer for his positive comment